# Channel Reduction for an EEG-Based Authentication System While Performing Motor Movements

**DOI:** 10.3390/s22239156

**Published:** 2022-11-25

**Authors:** Ellen C. Ketola, Mikenzie Barankovich, Stephanie Schuckers, Aratrika Ray-Dowling, Daqing Hou, Masudul H. Imtiaz

**Affiliations:** Department of Electrical and Computer Engineering, Clarkson University, Potsdam, NY 13699, USA

**Keywords:** authentication, biometrics, channel reduction, electroencephalogram, machine learning, random forest

## Abstract

Commercial use of biometric authentication is becoming increasingly popular, which has sparked the development of EEG-based authentication. To stimulate the brain and capture characteristic brain signals, these systems generally require the user to perform specific activities such as deeply concentrating on an image, mental activity, visual counting, etc. This study investigates whether effective authentication would be feasible for users tasked with a minimal daily activity such as lifting a tiny object. With this novel protocol, the minimum number of EEG electrodes (channels) with the highest performance (ranked) was identified to improve user comfort and acceptance over traditional 32–64 electrode-based EEG systems while also reducing the load of real-time data processing. For this proof of concept, a public dataset was employed, which contains 32 channels of EEG data from 12 participants performing a motor task without intent for authentication. The data was filtered into five frequency bands, and 12 different features were extracted to train a random forest-based machine learning model. All channels were ranked according to Gini Impurity. It was found that only 14 channels are required to perform authentication when EEG data is filtered into the Gamma sub-band within a 1% accuracy of using 32-channels. This analysis will allow (a) the design of a custom headset with 14 electrodes clustered over the frontal and occipital lobe of the brain, (b) a reduction in data collection difficulty while performing authentication, (c) minimizing dataset size to allow real-time authentication while maintaining reasonable performance, and (d) an API for use in ranking authentication performance in different headsets and tasks.

## 1. Introduction

Sensitive personal data is often stored on digitized databases that can be obtained through online portals using internet access. Traditionally, sensitive data has been protected using passwords or physical access keys [1,2]. In the advent of increased accessibility to biometric scanning tools in consumer devices, multiple data-storage platforms have adopted fingerprint and facial recognition as an additional security feature [3]. Biometrics are a secure recognition method that uses unique biological properties of an individual to ensure correct authentication occurs [4]. For example, fingerprints, iris, and facial recognition all display unique attributes of an individual [5]. In recent studies [6,7,8,9,10,11,12,13,14,15,16] electroencephalographic (EEG) signals were tested for their use in this biometric domain. It was postulated that EEG can serve as a biometric method for person authentication due to the unique pathways developed by each individual to accomplish a given mental activity [17].

EEG authentication systems rely on the ability to collect and analyze brain signals to match an individual to known activity templates. These systems require a data acquisition device, as well as a task for the user to perform. Data acquisition is typically accomplished by a wearable headset with electrodes placed at specific locations [14]. Users wear the headset while performing a given task, which stimulates the brain and allows the headset to record the electrical activity in the electrodes. Previous work has included tasks such as imagined hand movement [8,11,15,18], resting [8,9,10,12,13,14,16], limb movement [8,11], and visual tasks [6,19]. The chosen task(s) should be engaging enough to maintain user focus, but also simple enough to be performed without extensive instructions and training. The authentication system proposed in this study uses lifting small objects as a task, which meets these ideals of stimulating yet simple.

Several groups developing EEG authentication systems have used the EMOTIV Epoc+ EEG headset, which contains 16 electrodes: 14 EEG channels and 2 reference electrodes [6,9,14,15]. Other EEG headsets include the 64-electrode BCI2000 as used by [8,12], 19-channel TruScan [11], and single channel Mindwave [13]. The electrode placements are typically selected from the international 10–10 system shown in Figure 1. The international 10–10 system creates a schema by which 81 locations on the scalp are named and marked at 10% increments between the nasion (top of the nose) to inion (protuberance on the back of the skull), and left to right preauricular points (top of ear base) [20]. This is an extension of the original 10–20 system used to mark 21 locations on the scalp such that studies collecting EEG data can be standardized and reproducible. Newer systems such as such as the 128-electrode Quik-Cap, Waveguard, and actiCap or 256-electrode BrainAmp, Quik-Cap, and Electro-Cap use the newer 10–5 system with 345 standardized positions at 5% intervals [21]. The trend of increasing the number of electrodes improves the spatial resolution and accuracy of EEG data [20,22], however increasing the number of data streams also increases the size and processing time of EEG data.

Another factor that increases the data size is the sampling rate at which EEG data is collected. For example, the BrainID system used a sampling rate of 128 Hz while users were tasked with thinking about a specific four-digit code [14]. Since each of the 14 electrodes collect 128 samples a second, a total of 1792 samples are obtained per second by the system. Many recent studies employ 32–128 electrodes for data collection with sampling rates ranging from 128 Hz to 2400 Hz [23]. In systems with higher sampling rates and large numbers of electrodes, the size of the EEG data can grow exponentially. For instance, a standard 16-bit resolution 64-electrode system sampling at 512 Hz over each electrode will produce 32,768 samples each second. This evaluates to 524,288 bits or 65.536 kb or data that must be stored and processed for each second of a recording.

Larger datasets provide more detailed information but also increase the computation time needed to authenticate individuals using current machine learning models, resulting in multiple hours of processing time. For EEG-based authentication to shift from an area of theoretical research to consumer biometric, systems will have to authenticate individuals in real-time with minimal delay. To counter longer processing times, devices with increasingly larger memory can be used as the size of EEG datasets become more inflated. Even with high memory devices, EEG data processing time can be brought down by reducing the number of channels in consideration in the authentication process. To perform a reduction in the number of channels, it is essential to analyze each channel’s contribution to authentication performance. An ablation study can be performed to observe the contribution of each channel towards the overall classification results. This is performed by removing the channels with the least contribution in a sequential fashion and finding the lowest number of channels needed to maintain acceptable authentication performance.

Depending on the user task selected for authentication, different frequencies of the brain may provide the most synchronized, and therefore detectable, signals. Brain activity can be divided into five sub-bands: delta (0.5–4 Hz), theta (4–8 Hz), alpha (8–12 Hz), beta (12–30 Hz), and gamma bands (30 Hz and higher) [24]. Each sub-band is typically associated with specific functionalities of the brain and are typically active over specific regions of the brain, thus are detected by certain electrode positions with better precision than others. The delta and theta sub-bands are detected most strongly in the frontocentral region of the brain during sleep [25]. Alpha sub-band rhythms are observed in relaxed individuals in the occipital region of the brain, typically while the eyes are closed and visual stimuli is limited [26]. When attentive to stimuli or problem-solving, the beta sub-band is active in the frontal and temporal regions for short periods of time [27]. Gamma sub-bands are seen during movement, emotional processing, and high-level mental activities and can be observed over multiple regions of the brain [26,28,29]. All five sub-bands are analyzed alongside the non-segmented EEG data to observe changes to EEG authentication performance in this study.

In summary, the contribution of our study is: (a) To evaluate EEG as a biometric authentication procedure when users perform a low-impact lifting task. (b) Determining the lowest number of electrodes required to perform authentication with acceptable accuracy. (c) Find the best electrode positions and frequency band for this authentication task for the design of a specialized authentication headset. (d) Share an open-source repository with a channel reduction and sub-band analysis API.

## 2. Materials and Methods

### 2.1. Dataset

This study utilizes a publicly available WAY_EEG_GAL dataset from Luciw et al. [30] that was collected without intent for use in user authentication. The WAY_EEG_GAL dataset [30] is made up of 3913 grasp and lift trials from 12 participants (8 female, 4 male). Each participant performed 328 trials in which they lifted an object of either 165 g, 330 g, or 660 g from a contact surfaces of either silk, suede, or sandpaper using their thumb and index finger. The combination of these two variables were randomly ordered during data collection. During trials, the object was lifted and suspended until an LED light cued the participant to place the object back down, therefore the time of each trial varies.

The dataset was recorded across kinematics, force, torque, EEG, and electromyography (EMG) modalities. Four 3D position sensors were attached to the item that was picked up by the participant to track movement, and two contact plates on the side of the item collected 3 channels of force and torque data to detect grip. The contact plates are located on the Fz plane shown in Figure 2a. An ActiCap headset with 32 channels at a sampling rate of 500 Hz was used to collect EEG data as seen in Figure 2c. The third image in Figure 2b showcases five sensors on the anterior deltoid, brachioradial, flexor digitorum, common extensor digitorum, and the first dorsal interosseus muscles which were used to collect EMG data. Of the available modalities, only EEG is utilized in this study.

### 2.2. Experimental Setup

The experimental setup used to preprocess the EEG data and implement the machine learning model for user authentication and channel reduction is described in this section to allow for experimental replication. The experimental setup also specifies the system requirements for using the API associated with this study. The API only works on machines that have access to a NVIDIA GPU and Linux OS. A GPU is utilized due to the ability to run multi-thread instructions that significantly speed up the machine learning training and classification process over traditional CPU-based computing. The following system setup variables were used with the API in this study:
**Operating** **System.** 
Linux (Ubuntu 18.04, Ubuntu 20.04, CentOS 7, CentOS 8, RHEL 7&8).
**Environments.** MATLAB R2021b and Python 3.9 (Conda RAPIDS-22.02).
**Dependencies.** (Matlab) Signal Processing Toolbox, (Python) NVIDIA CUDA 11.5, cuML, sklearn, imblearn, numpy.
**Minimum** **Suggested** **Memory.** 
16 GB DDR4 RAM with 2 GB SWAP.


### 2.3. Data Pipeline

The following data processing pipeline was performed in this study:

#### 2.3.1. Filtering

The EEG data is filtered using a MATLAB script. A 4th-order IIR Butterworth filter from the Signal Processing Toolbox is applied to the imported EEG data with a pass-band of 0.2–50 Hz. The band-pass frequencies are converted into a normalized frequency, $W_{n}$ using the following formula:$$W_{n}=\frac{f_{cutoff}}{\frac{f_{s}}{2}},$$
where $f_{cutoff}$ is the low or high pass-band, and $f_{s}$ is the sampling rate of the data. The normalized frequency of the high-pass and low-pass frequencies are applied to the 4th-order Butterworth filter. The Butterworth filter Matlab execution can also be found in the Preprocessing folder of the Github repository [31]. The Butterworth filter is applied to the EEG data using the filtfilt function. This function was chosen due to compatibility with zero-phase lag IIR filters.

#### 2.3.2. Band Extraction

After all filtering is completed, the filtered EEG data is segmented into five frequency bands using a MATLAB script. The extracted bands are delta (0.2–4 Hz), theta (4–8 Hz), alpha (8–12 Hz), beta (12–26 Hz), and gamma (26–50 Hz). Each band is extracted using a 4th-order Butterworth filter in the same manner as described in Section 2.3.1, with the exception of the delta band which is extracted using a 2nd-order Butterworth filter. The filter order is decreased in the delta band due to the 0.2 Hz high-pass band being below the recommended 1 Hz minimum for 3rd-order and higher IIR filters [32]. A collective band without any extracted frequencies and the five sub-bands results in a total of 6 bands to be classified. Each band is preprocessed and classified independently in the remainder of the data pipeline.

#### 2.3.3. Windowing

The EEG data from each band is segmented into overlapping rectangular windows in MATLAB. Each window contains 0.5 s of EEG data, with a 0.25 s overlap with previous and future windows. Windowing is used to isolate features over smaller segments of the overall EEG data to improve feature resolution [33].

#### 2.3.4. Feature Extraction

Features are extracted from each band of the EEG dataset to provide a concise representation of the EEG signals using information that is useful to the classifier model. Each feature describes a property of the signal contained within each window. A set of 12 statistical features are extracted from each window. Because the size of each window is small in comparison to the entire dataset, computing only 12 properties to represent the entire window is unlikely to result in significant data loss due to the data reconstruction using all windows combined.

Each of the 12 extracted features were selected from current literature in EEG user authentication [34,35]. The 12 features used are average, standard deviation, mean absolute value, root mean square, skewness, kurtosis, Hjorth activity, Hjorth mobility, Hjorth complexity, Shannon’s entropy [36], and spectral entropy [37].

All features are obtained using Matlab functions. Statistical features such as average, standard deviation, mean absolute value, root mean square, skewness, and kurtosis are obtained from the time-domain data to provide relationships between the samples within individual windows. The Hjorth parameters are found using the Hjorth function from the MATS toolkit [38]. Hjorth parameters are advanced statistical methods in observing time-domain signals through patterns in variance, making this a popular method in monitoring EEG data [39]. The entropy features describe the amount of uncertainty in the EEG signal pattern. This is observed in both the time domain using Shannon’s entropy and in the frequency domain using spectral density. Both entropy features are obtained by applying the windowed data using modified wavelet entropy functions.

#### 2.3.5. Data Splitting

The six bands for each participant were then split to create training, validation, and testing data. For each feature dataset, 80% was used for training, 10% for validation, and 10% for testing. Samples were randomly selected from each featureset while splitting to avoid time-bias. All data processing performed after this step are completed in Python using the GPU-dependent API.

#### 2.3.6. Channel Ranking

Since 32 channels of data are available, testing every non-ordered combination of possible electrodes within the headset would result in 32 choose k models being needed for testing. This evaluates to
C32,k=32k=32!k!(32−k)!
$$C_{32,k}=4294967296\;models\;required.$$

Since this is a large number of models to run on a consumer-grade computer, a channel ranking system is used to only evaluate 32 possible combinations with the total number of channels decreasing in each permutation. Channel ranking is performed using the Extreme Gradient Classifier (XGB) model from python’s xgb module. This model is an ensemble class algorithm. This means that the amount of inter-subject variance accounted for by each channel towards the entire dataset is scored as a percentage. A percentage is returned for each channel as an unsorted array. This keeps the percentage for each channel in the same column as the channel it is representing.

These channel rankings are used for channel reduction by removing the channel with the lowest percentage from the binerized dataset across both training and testing data splits. Both the testing and training data are reduced to simulate the use of a headset with the chosen channel’s electrode removed. This reduced dataset is used to retrain the random forest model using the training split. The retrained model is used to predict the classes of the reduced testing data. The physical location of each channel which is removed is kept on record to analyze the patterns in channel importance.

#### 2.3.7. Upsampling

User authentication is performed by comparing the templates of one single user to the templates of all 12 users on the database. The code will select one participant as a ‘genuine’ user and will mark all other users as ‘imposters’. The code cycles through each participant such that each individual will be the genuine user once and the imposter 11 times. To simplify the identification of genuine user versus imposter, all training data is binarized, while a user is selected as the genuine user, all training samples labeled with the genuine user’s ID have their labels changed to a ‘1’, while all other user IDs are changed to a ‘0’.

However, training a machine learning model with a 1:11 ratio of genuine samples to imposter samples will result in classification biases towards imposter classifications. To circumnavigate this potential issue, the number of genuine samples are increased to match the number of imposter samples to achieve a 50/50 split in genuine and imposter training data. This upsampling process is performed using Synthetic Minority Over-sampling Technique (SMOTE) [40].

SMOTE locates the minority class in a dataset, which in the case of this study will be the genuine user’s data. Given that binary labeling is used, the SMOTE algorithm will identify the discrepancy between the number of available genuine samples to imposter samples by finding the ratio of ‘1’ labels to ‘0’ labels, while the genuine samples remain at a lower ratio, new samples will be created. These new samples are made by selecting the k nearest neighbors and creating a convex combination of two neighbors [40].

#### 2.3.8. Hyperparameter Tuning

The binarized and upsampled data is used to train and fit a random forest model. A benchmark is created for the random forest model in which all 32 channels of data are used in fitting and classification. The random forest model is tuned by changing the values of the hyperparameters, after which the model is trained and tested. The best hyperparameter combination is determined and used in all further models trained when channel reduction is performed to ensure results can be attributed to channel reduction exclusively. The chosen hyperparameter combination will be used to generate the 32-channel benchmark score that is used for comparison with channel reduction scores.

Hyperparameter selection is performed in an iterative loop. For hyperparameters with a finite number of predetermined options, a list of possible values are made. For hyperparameters with infinite options of numerical values, a range of numbers are placed in a list. The numbers for each numerical hyperparameter were determined by first testing a broad range of values that varied from 10% to 1000% of the module’s default value. After testing across all 12 participants, the range is narrowed down until different values provide optimal results for different participants.

The random forest model used in this section is obtained from the CUDA cuML ensemble library. The cuML ensemble repository is made to reflect the structure of sklearn’s ensemble library, and thus the functionality of this code can be changed to run on CPU exclusive devices. Changing from the cuML to sklearn ensemble libraries can result in improved classification performance due to constraints on GPU machine learning kernels, however making this change will also result in a much greater processing time which is not ideal for real-use in-person authentication systems.

#### 2.3.9. Classification

Classification is performed using the same random forest parameters specified in the benchmark test in a model trained using channel reduced datasets. For each lowest ranking channel that is truncated from the dataset, the change in classification accuracy is saved to a results file. The accuracy of the classifier is determined by the percentage of times the model is able to correctly classify a 0.5 s sample across all available channels as being genuine or imposter. The results file contains the benchmark score, impurity score list, training and classification times, and accuracies for each channel reduction.

#### 2.3.10. Analysis

An analysis API was developed to interpret and plot specific patterns in the results. This API was developed in Python 3.9 as a class with functions to perform different types of analysis procedures on the results file output by the classification script. The API can be imported into scripts to use the averaging, data sorting by participant, sorting data by band, and timing value extraction functions. These values are returned as lists that can be input to matlibplot pyplot objects for plotting, or to be used for further user-defined analysis. The following analysis operations were applied to the classification results:Individual participant accuracy.-The classification accuracy for each participant is plotted against each other at each channel reduction for each of the 6 bands;-This allows for discrepancies between participant performance to be viewed in each band.Band performance by participant.-The accuracy at each number of channels used is plotted for each participant across all bands;-This allows for differences in band performance to be observed for each participant.Average band accuracy.-The average accuracy over all participants for each number of channels used is calculated for each band;-The average accuracy for each band is plotted against each other to view differences in average band performances.Participant Gini importance rankings.-The Gini importance assigned to each electrode position is extracted for each participant and plotted on a bar graph.Average Gini importance by band.-The average Gini importance of each electrode position is calculated across all participants; by band;-A bar graph is made to visualize the average ranking of electrode positions for each band.Change in accuracy.-The change in accuracy for each channel reduction is found for each participant in each band;-The changes for each participant are plotted against each other within each band to determine if change per participant is significantly different.Average classification time by band.-The average classification time across all participants for each number of channels used for each band is determined;-The average classification time of each band is plotted against each other to review differences in band timing performance.Channels reduced before 1% reduction in accuracy.-The number of channels that can be reduced before the average accuracy is reduced by 1% is found in each band;-This finds the minimum channels needed before the accuracy is dropped by 1% in each band.

All plots are graphed using matlibplot’s pyplot.

## 3. Results

When observing all six of the examined bands, it can be seen that the bands follow trends in performance across participants. Figure 3 showcases the average performance of each band by number of channels used. This average was obtained by finding the mean accuracy across all 12 participants in a given band for each number of channels used. From the average performance, it can be seen that the gamma sub-band provides the highest authentication accuracy at 83.15%, followed by the beta sub-band and overall band at 81.95% and 81.35%, respectively. The worst performing bands are delta and theta closely tied together at 78.13% and 78.46%, respectively, followed by alpha at 79.85%.

The change in accuracy per channel reduction is also viewed by individual participant over all six bands. This process is performed to ensure that the average accuracy provides a consistent representation across all individuals, and to view the degree of variation observed across individuals. It is observed that while gamma provides the highest average performance, it is not the top performing band in all participants, as seen in Figure 4 where the overall and beta bands outperform gamma. However, the gamma, beta, and the overall bands are the top performing bands in the vast majority of participants, thus excluding the theta, delta, and alpha bands as significant contributors. Within the three top performing bands, gamma is most often seen as the top performance band, as seen in Figure 5 and Figure 6.

When assessing accuracy, it can be seen that not all participants achieve similar accuracies. A higher authentication accuracy is observed for participant 2 than with all other subjects. Similarly, participant 10 is not authenticated at as high a rate as as other participants across all six bands. A significant decrease in accuracy of ∼10% can be observed in specific bands between these two participants.

To determine the optimal number of channels, the change in accuracy for each channel reduction is observed. Figure 7 showcases the change in accuracy seen for each participant in the overall band. The decrease in accuracy when 1–20 channels are removed remains minuscule; however when considered alongside the average accuracy in Figure 3 it can be seen that large decreases in accuracy occur only at below 10 to 7-channels. The minimum number of channels needed before a 1% accuracy reduction occurred was determined by band. The minimum channels required is 13 for the gamma sub-band, 14 for the overall band and beta sub-band, 15 for the theta and alpha sub-bands, and 17 for the delta sub-band.

To determine the best electrodes for use in a motor task-based authentication system, the channels assigned the highest Gini importance during channel ranking are considered. The Gini importance of each channel varies depending on the band selected. The best electrodes for use in this system are therefore selected from the gamma, beta, and overall bands. The average Gini importance of each electrode is obtained by using the mean value across all participants in a band. Figure 8 shows the average Gini importance of each electrode in the gamma band. The top 14 channels can be seen in Table 1. The physical location of each electrode on the international 10–10 electrode placement system can be seen in Figure 1.

No linear relationship exists between the number of channels reduced and the user authentication time, however a notable pattern exists among all the sub-bands observed. The exclusion of the worst performing channel results in a noticeable decrease in average classification time in all bands. After the first channel is removed, the classification time exhibits a gradually escalating upward trend until only 7 to 10 electrodes are left. All bands experience the longest classification time at six electrodes, at which point further channel reductions result in a steep decline in classification time. The average classification time during channel reduction exceeds the average classification time for all 32 channels at 23 channels for the beta sub-band, 18 channels for the overall band, gamma and theta sub-bands, 17 channels for the delta sub-band, and 16 channels for the alpha sub-band. Figure 9 showcases these findings. The time required to train each model does see a linear trend as channels are reduced.

## 4. Discussion

This study aims to reduce the computational intensity and storage demand required for EEG-based user authentication by minimizing the number of electrodes in use. By finding the optimal number and location of electrodes, a custom headset can be designed to perform this task while reducing user discomfort due to bulky EEG headset designs. This headset would be specialized for the recognition of motor-based tasks such as the one performed by participants in this study for use in instant authentication. The minimum number of channels that can be used while maintaining an accuracy drop-off of less than 1% from using all 32 channels ranges from 13 channels to 17 channels across the different bands analysed. This indicates that the number of electrodes in use for the specialized EEG headset should fall within this range. As seen in Figure 3, the gamma, beta, and overall bands provide the highest accuracy of the six available options, indicating that the number of electrodes if using the best performing bands can be reduced to either 13 or 14 electrodes. Therefore, a 14-channel headset is proposed for collecting enough data to accurately perform user authentication while minimizing the size of the dataset passed to the machine learning algorithm. The proposed headset will utilize electrodes placed at the locations corresponding to the top 14 channels. The overall top 14 channels are considered those that were listed among the top 14 by individual participants the most often. From the data listed in Table 1, the top 14 channels are ‘PO10’, ‘T8’, ‘F4’, ‘O2’, ‘Fp2’, ‘F3’, ‘T7’, ‘PO9’, ‘FC5’, ‘F7’, ‘O1’, ‘Fp1’, ‘Oz’, and ‘Fz’. These top 14 electrodes have the highest Gini importance score across the gamma, beta and overall bands. The location of these electrodes can be seen in Figure 10. It should be noted that half of the 14 top performing channels are clustered around the region of the frontal cortex. These results make sense given the task for the dataset used, as the frontal cortex is known to be heavily connected to motor movement [41,42]. In addition to the frontal lobe positioned electrodes, 5 of the top ranked electrodes are located above the occipital lobe. The occipital lobe is associated with the processing and interpretation of visual information [43]. Fine motor movement tasks, such as the task performed by participants in the public dataset used, utilize vision for motor planning and spatial perception [44]. As such, the high performance of the occipital lobe sensors lines up with the known functions of the brain regions. It can also be noted that two symmetric temporal lobe electrodes are included in the top ranking electrodes. The temporal lobe is yet another region involved in the process of spatial recognition and thus it’s inclusion in a headset may improve fine motor movement recognition [45].

Creating a 14-electrode authentication EEG headset will allow for increased user comfort, as most EEG collection devices require close contact of the electrodes and scalp. As a result, an increased number of electrodes involves greater effort from the headset wearer to part hair to provide access to the scalp. In addition, many EEG devices require the use of saline solution or even electroconductive gel. By reducing the number of locations where these solvents are applied, the user discomfort can be minimized. The top ranked electrodes are evenly mirrored across he left and right hemispheres of the brain, with the exception of FC5 and F7, indicating that a well balanced headset could be obtained for this functionality. Further research may be conducted to analyze exchanging the FC5 electrode for the F8 electrode, or exchanging the F7 electrode for the FC6 electrode to maintain even electrode symmetry.

From a cost standpoint, reducing the number of electrodes in use by an EEG headset should allow for the development of cheaper EEG authentication systems. Less electrodes and connecting materials such as wires, plastic, and insulation are required in the design, which allows for a reduced manufacturing cost. In addition to mechanical materials, the embedded components of such a system will also be available at a reduced price. Less channels involves the use of less analog to digital converters needed for microcontrollers to interpret the EEG signals. This reduces component prices for the design, but also allows for the use of more general purpose embedded devices that do not need specialized additions to accommodate for large numbers of analog inputs. The use of general embedded devices widens the range of devices open to be interfaced with the proposed headset, and thus increases the ability to perform further improvements by independent researchers at a low cost.

The computational power required to process the EEG signals from a 14-channel headset is significantly lower than that required for a 32-channel or larger headset. Since the processing power of computers are limited by memory, increased data sizes can lead to the memory cap being reached and therefore may exponentially increase computation time once said cap has been reached. To allow low-cost devices with low memory to perform authentication tasks, reducing the strain on system memory provides great benefits to processing time. This allows for real-time authentication to be performed on pre-trained models.

The timing results obtained in this study indicate that reducing the number of channels in use does not result in large decreases in classification time when using the RF classifier, and that classification time can increase when large numbers of channels are removed. For the 14-electrode system proposed, the classification time remains similar to that of a 32-electrode system. The initial drop in classification time when the first channel is removed may be due to reduced dimentionality paired with simplification from reduced overfitting [46]. Subsequent reductions after 23–16 channels may increase the classification time because smaller datasets may lead to less effective use of parallelization. Only the significantly reduced number of channels decreases the time needed for classification, indicating there may be a threshold for effective tree parallelism. The training time for the model is linearly reduced, and as a result enrollment time could be reduced using a reduced channel system. The accuracy score of 82.25% with 14 electrodes using the gamma sub-band shows promising results for the proposed EEG headset design. This acts as a proof of concept that a motor task EEG-based user authentication system could be implemented at a wider scale with future research for improving the model. Many changes that may improve the accuracy even further include investigating the use of different features, feature fusion, and the use of a feature selection algorithm as part of preprocessing, the use of different machine learning models for comparison of results, comparing different channel ranking models for changes in authentication accuracy, and investigating mutual information between channels on a participant by participant basis. Additional methods that may improve the performance includes the use of multimodal data for user authentication, and observing band combinations.

The public dataset was employed in this study was not collected for authentication. As a result, the performance obtained by authenticating users from this dataset may be lower than if a dedicated dataset intended for authentication use where employed. Factors such as task times, types, and labeling may be modified in a dedicated dataset to result in higher authentication accuracy. Further investigation into the proposed design may be conducted through the use of self collected EEG data with users instructed to perform motor-tasks for user authentication.

An additional consideration when analyzing the results of this paper is the size of the userbase. The public database only contains 12 participants. This small pool of participants may not provide a widely applicable conclusion to the brain activity of fine motor tasks across the wider population of individuals. Replicating the study with a larger userbase will greatly improve the reliability of the results.

Future work investigating the connection between top performing channels and types of tasks may be beneficial. It’s believed that visual stimuli are able to produce more unique EEG data [9]. Therefore, an analysis of the performance of a variety of electrodes when complemented by a visual task may produce interesting results.

In summary, the study provides a proof of concept for a custom motor-task EEG-based authentication headset for use in real-time user authentication. This headset will involve the use of 14 electrodes with a focus over the frontal cortex and occipital lobe. The reduced data size of this headset will reduce the computational complexity while only seeing a 1% drop in authentication accuracy compared to the original 32 channel headset the data was collected with.

## 5. Conclusions

In this project it was found that 32 channels of EEG data can be truncated down to 14 channels while maintaining authentication performance within a 1% accuracy decline of the full channel system when users perform a simple motor task. This is a reduction in over half the dataset, thereby decreasing the time required to (train and) predict if a user is genuine or an imposter. These results indicate that a custom headset with 14 electrodes clustered over the frontal lobe, occipital lobe, and temporal lobe of the brain can be designed for a motor movement task-based authentication system. This may reduce data collection difficulty and reduce dataset size while maintaining accuracy performance.

## Figures and Tables

**Figure 1 sensors-22-09156-f001:**
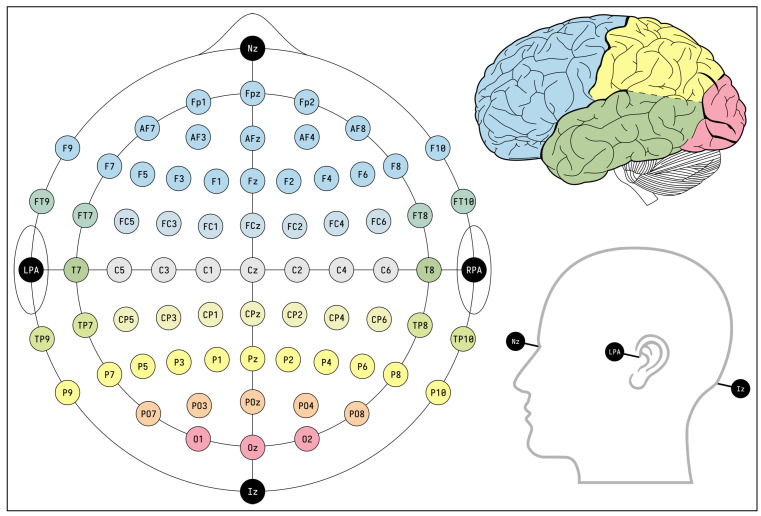
Normal placement of electrodes according to the international 10-10 system.

**Figure 2 sensors-22-09156-f002:**
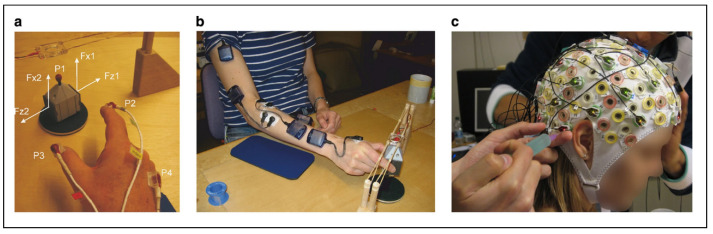
The data collection procedure for the EEG_WAY_GAL dataset. (**a**) Force detection sensors. (**b**) EMG sensor placement. (**c**) ActiCap in use for EEG data collection [30].

**Figure 3 sensors-22-09156-f003:**
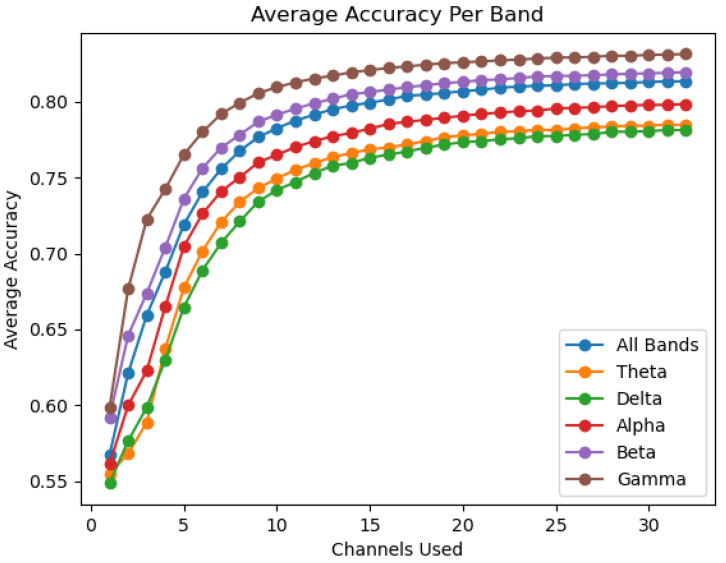
The average accuracy obtained by each band for each number of electrodes in use.

**Figure 4 sensors-22-09156-f004:**
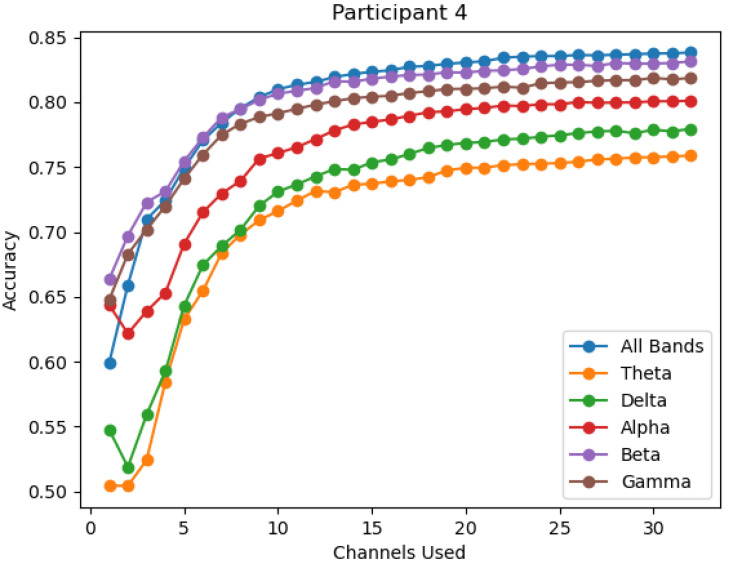
The accuracy of each band for participant 4.

**Figure 5 sensors-22-09156-f005:**
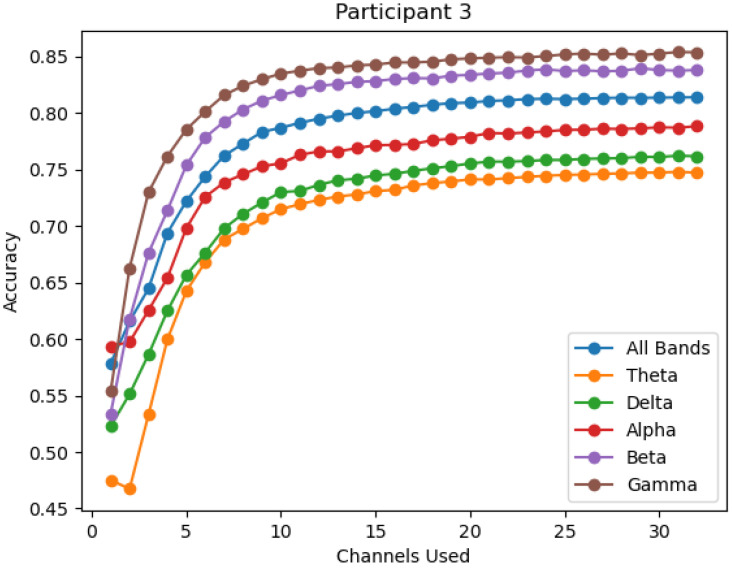
The accuracy of each band for participant 3.

**Figure 6 sensors-22-09156-f006:**
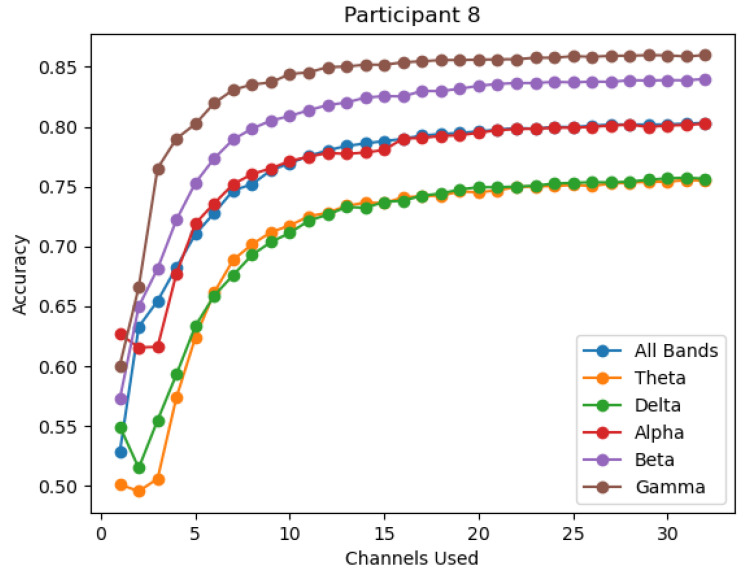
The accuracy of each band for participant 8.

**Figure 7 sensors-22-09156-f007:**
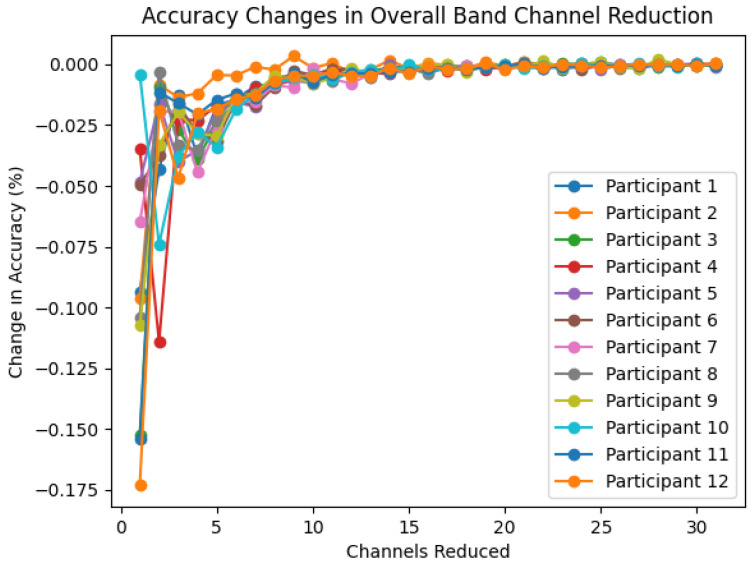
The change in accuracy for each participant using the overall band for each number of channels used.

**Figure 8 sensors-22-09156-f008:**
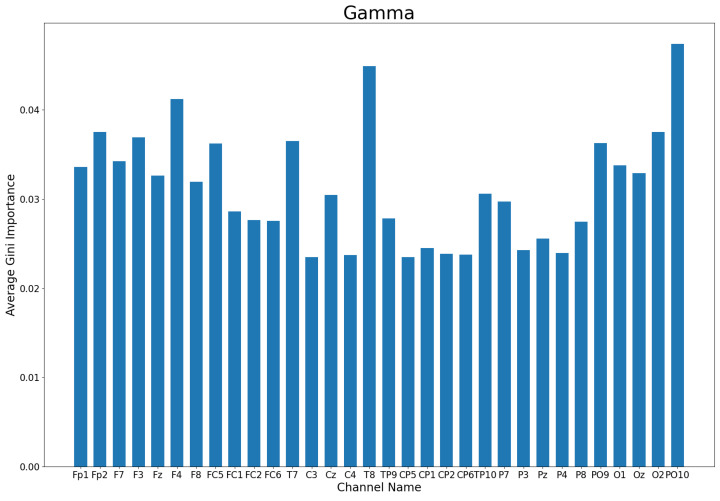
The average Gini importance for each electrode in the Gamma band.

**Figure 9 sensors-22-09156-f009:**
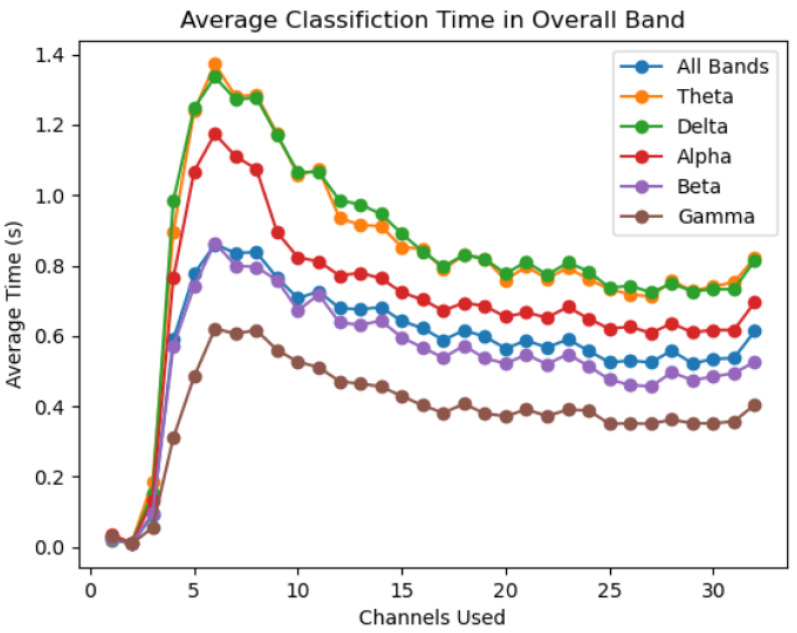
The average classification time for each number of channels used in each band.

**Figure 10 sensors-22-09156-f010:**
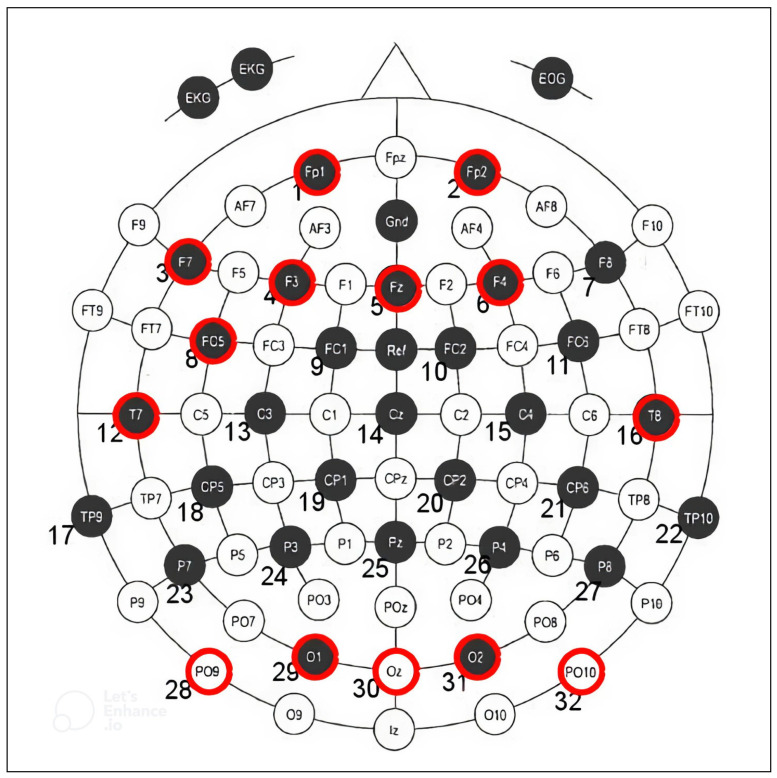
The placement of electrodes by channel number in the WAY_EEG_GAL dataset with the top 14 channels marked in red.

**Table 1 sensors-22-09156-t001:** Channels selected ranked in top 14 during channel ranking.

Channel	Fp1	Fp2	F7	F3	Fz	F4	F8	FC5
**Times in Top 14**	6	11	8	12	10	11	5	7
**Channel**	**FC1**	**FC2**	**FC6**	**T7**	**C3**	**Cz**	**C4**	**T8**
**Times in Top 14**	9	5	4	7	3	10	3	9
**Channel**	**Tp9**	**CP5**	**CP1**	**CP2**	**CP6**	**TP10**	**P7**	**P3**
**Times in Top 14**	1	3	2	1	4	0	2	0
**Channel**	**Pz**	**P4**	**P8**	**PO9**	**O1**	**Oz**	**O2**	**PO10**
**Times in Top 14**	5	0	2	5	3	6	3	11

## Data Availability

The data presented in this study are openly available in the EEG_User_Auth repository from user KetolaC on Github.

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
