# Peer review of "Channel Reduction for an EEG-Based Authentication System While Performing Motor Movements"

_sensors, 2022, doi:10.3390/s22239156_

Round 1

Reviewer 1 Report

To improve the paper further, I have the following suggestions: 1. In Section 2.3 Pipeline, authors should give more details about these steps: (1) in subsection 2.3.1, what are the specific parameters of the IIR Butterworth filter and the pass-band frequencies used? These parameters should be shown in the paper, rather than asking for readers analyzing the code in github repo to find them. (2) in subsections 2.3.8 and 2.3.9, how to tune the random forest model is not clear, and the method changing the hyperparameters is also not given in the paper. (3) in subsection 2.3.10, what is the method or procedure used in the “analysis API developed”?  2. Authors can investigate the results raised by points 30-32 in Figure 9, and give reasonable description. It is not convincing enough based on the illustration in Page 8. 3. In page 8, section 4, authors claim that “only 14 electrodes are needed to maintain acceptable performance”. Please give the specific thresholds used to define “acceptable performance”. 4. Figure 10 is from reference [39] originally. It cannot be taken into this paper without any changes. Authors can delete it. As for “the functions of the brain lobes”, authors can illustrate using their own words.  5. More papers published in this year 2022 should be reviewed in section 1 Introduction.

6. I suggest authors investigate results raised by points 30-32 in figure 9,and give a reason. It is not convincing enough based on the illustration in page 8.

Author Response

Hello,
Please see the attachment.
Best Regards,

Ellen Ketola

Reviewer 2 Report

The topic is very interesting when it comes to the application of EEG signals in authentication services.

1. It should be kept in mind that the EEG source is a biometric source and for these reasons I would suggest the authors to have more fun with the information analysis of the biometric source.

2. For example, one feature "Shannon entropy" was mentioned. I would like to see the detailed procedure of how channel entropy is determined?

3. Also, for the analysis of biometric sources, it is important to mention mutual information, I believe that it would be useful when choosing a channel. (Mutual information between different persons is expected, but we are interested in channels with high mutual information in one person)

4. I am glad that there are developments in this area, it should be remembered that the number of respondents is very small, regardless of the use of the synthetic method SMOTE.

5. I welcome the mentioned contributions, the reduced number of channels, I believe it will affect the ergonomics of hardware solutions as well as data processing.

6. It is not clear to me, when you say "reliable high accuracy was achieved with 14 channels", do you mean 82% accuracy. Explain?

7. In some paragraphs the continuous authentication system is mentioned, is the system intended for that purpose or for instant authentication services?
